# Comparison of Four Commercial Screening Assays for the Detection of bla_KPC_, bla_NDM_, bla_IMP_, bla_VIM,_ and bla_OXA48_ in Rectal Secretion Collected by Swabs

**DOI:** 10.3390/microorganisms7120704

**Published:** 2019-12-16

**Authors:** Francesca Del Bianco, Manuela Morotti, Silvia Zannoli, Giorgio Dirani, Michela Fantini, Maria Federica Pedna, Patrizia Farabegoli, Vittorio Sambri

**Affiliations:** 1Unit of Microbiology, The Great Romagna Hub Laboratory, 47822 Pievesestina (FC), Italy; manuela.morotti@auslromagna.it (M.M.); silvia.zannoli@auslromagna.it (S.Z.); giorgio.dirani@auslromagna.it (G.D.); michela.fantini@auslromagna.it (M.F.); mariafederica.pedna@auslromagna.it (M.F.P.); patrizia.farabegoli@auslromagna.it (P.F.); vittorio.sambri@auslromagna.it (V.S.); 2Department of Experimental, Diagnostic and Specialty Medicine, University of Bologna, 40126 Bologna, Italy

**Keywords:** Enterobacteriaceae, multidrug-resistant organisms, asymptomatic colonization, screening assays

## Abstract

The spread of carbapenem-resistant Enterobacteriaceae (CRE) has been enabled by the lack of control measures directed at carriers of multidrug-resistant organisms in healthcare settings. Screening patients for asymptomatic colonization on the one hand, and implementation of contact precautions on the other hand, reduces patient-to-patient transmission. Screening plates represents a relatively low-cost method for isolating CRE from rectal swabs; however, molecular assays have become widely available. This study compared the performance of four commercial molecular platforms in detecting clinically significant carbapenemase genes versus routine screening for CRE. A total of 1015 non-duplicated rectal swabs were cultured on a chromogenic carbapenem-resistant selective medium. All growing Enterobacteriaceae strains were tested for carbapenemase-related genes. The same specimens were processed using the following molecular assays: Allplex™ Entero-DR, Amplidiag^®^ CarbaR + MCR, AusDiagnostics MT CRE EU, and EasyScreen™ ESBL/CPO. The prevalence of *Klebsiella pneumoniae* carbapenemase (KPC)-producing Enterobacteriaceae detected by swab culture was 2.2%, while organisms producing oxacillinase (OXA)-48 and metallo-β-lactamases were infrequent. The cost of CRE-related infection control precautions, which must be kept in place while waiting for screening results, are significant, so the molecular tests could become cost-competitive, especially when the turnaround time is decreased dramatically. Molecular assays represent a powerful diagnostic tool as they allow the rapid detection of the most clinically relevant carbapenemases.

## 1. Introduction

Antimicrobial resistance is one of the most complex global health challenges [1]. The World Health Organization puts the development of new antibacterial agents to treat carbapenem-resistant Enterobacteriaceae (CRE) among the most critical priorities [2]. Because CRE are resistant to the majority of β-lactams, carbapenem resistance has minimized the usefulness of many commercially available drugs [3]. In addition to that, CRE frequently carry mechanisms conferring resistance to other antimicrobial classes, thus further limiting the available therapeutic options [4,5,6]. Resistance to carbapenems is typically based on two main mechanisms. The first one is related to structural mutations combined with the activity of other β-lactamases, such as AmpC cephalosporinase (AmpC) and extended spectrum β-lactamases (ESβLs). The second, and likely the most important, mechanism is carbapenemase enzyme production. These versatile β-lactamases are able to hydrolyze carbapenems and other β-lactam antibiotics [7,8,9,10].

Early reports have described carbapenemases as chromosomally encoded β-lactamases that hydrolyze penems and carbapenems. The properties and characteristics of these enzymes are distinctive and species-specific [11,12,13].

Plasmid-borne carbapenemases are encoded by genes that are horizontally transferable. This group includes members of three out of the four β-lactamase classes (A, B, and D), according to the Ambler classification [14]. Class A and D carbapenemases require serine in the active site. Class B enzymes require zinc for β-lactam hydrolysis; therefore, they are renamed metallo-β-lactamases (MBLs). Class A KPC (*Klebsiella pneumoniae* carbapenemase) enzymes are currently the predominant carbapenemases reported worldwide among the Enterobacteriaceae [15]. KPC-producing *K. pneumoniae* is endemic in several countries, including the United States, Israel, Greece, and Italy [16]. VIM (Verona integron-encoded metallo-β-lactamase), IMP (active on imipenem), and NDM (New Delhi metallo-β-lactamase) are the more geographically widespread MBLs [17,18,19]. Regarding class D carbapenemases, OXA-48-like-carbapenemases are commonly widespread in enterobacterial species, especially in Turkey, Middle East, and North Africa. OXA-48-type enzymes show a high level of hydrolytic activity against penicillins and a low level of hydrolytic activity toward carbapenems [20,21].

The spread of carbapenemase-producing Enterobacteriaceae (CPE) has been sustained and favored by the combined effects of (i) the widespread use of carbapenems for the management of infections caused by ESβL-producing Enterobacteriaceae [22,23], (ii) clonal dissemination of particular clinically relevant strains [24], (iii) interspecies dispersion due to the wide variety of conjugative plasmids, which are well adapted to different bacteria [25,26,27], and (iv) co-selection of antibiotic resistance, as the use of one antibiotic is enough to maintain all drug resistance mechanisms since they are linked together on the same plasmid [28,29]. CPE are a common cause of healthcare-associated infections as well as colonization, which is a major source for the transmission of antimicrobial resistance-related genes inside and outside healthcare institutions [30]. From 2010 onwards, there was an increase in infections caused by *K. pneumoniae* resistant to carbapenems in the Emilia-Romagna region in northern Italy. To control the spread of CPE, a region-wide intervention was carried out based on a regional guideline issued in July 2011 and subsequently updated in February 2017 [31]. The strategies to reduce CPE transmission in healthcare settings are focused primarily on the early identification of infected patients and carriers and on the wide implementation of contact precautions. According to these current strategies, CPE infections and colonization are diagnosed based on phenotypic methods followed by phenotypic and/or genotypic confirmation. Additionally, rectal swabs are performed on asymptomatic carriers as a tool to achieve active surveillance that involves close contact with hospitalized patients with CPE infection, high-risk patients at hospital admission, and patients admitted to intensive care units, transplant units, oncology units, and hematology units. All CPE-infected patients and asymptomatic carriers should be subjected to contact isolation precautions (dedicated nursing staff and cohorting) during their stay in hospitals [32]. In the context of high CPE prevalence, rapid and accurate identification of colonized patients is essential for decreasing the diffusion of such microorganisms and to reduce the costs of all the unnecessary contact precautions [33].

This study evaluated the analytical performance of four commercial molecular technological platforms for the detection of the most clinically important carbapenemase genes in comparison with the culture-based routine laboratory method.

## 2. Materials and Methods

The study was conducted at the Microbiology Unit of the Greater Romagna Area Hub Laboratory, northeastern Italy, between May and July 2018. The Laboratory serves 1,250,000 inhabitants and 11 hospitals for a total of about 4200 beds.

A total of 1015 non-duplicated rectal swab specimens were prospectively collected using ESwab™ (COPAN Italia SpA., Brescia, Italy). The samples were transported to the Laboratory upon collection, processed within 24 h, and reported in 48 h.

### 2.1. Culture Based and Molecular Routine Methods for the Identification of CRE

Rectal swab specimens were collected as part of the Antimicrobial Resistance Surveillance Program, in which patients were screened for CRE carriage by rectal swab on admission, according to the hospital protocols. For the isolation of CRE strains from rectal swab specimens, a 10 μL aliquot from each specimen was inoculated into a CHROMID^®^ CARBA SMART selective chromogenic media bi-plate (bioMérieux, Marcy l’Etoile, France).

The plates were incubated at 37 °C for 18 h and then examined for growth (*Escherichia coli* showing dark pink or red colonies; *Klebsiella* spp., *Enterobacter* spp., and *Citrobacter* spp. showing metallic blue colonies).

If a swab tested positive, any suspected colony was further identified using the Vitek MS MALDI-ToF (bioMérieux, Marcy-l’Etoile, France).

Species identification was followed by a molecular test (Xpert^®^ Carba-R test, Cepheid, Sunnyvale, CA, USA) to detect the gene sequences from pure colonies. The following genes were identified: *bla_KPC_*, *bla_NDM_*, *bla_VIM_*, *bla_OXA-48_*, and *bla_IMP_*.

### 2.2. Molecular Tests

No ethical approval was required because only residual leftover specimens were tested with the evaluated molecular techniques, and patient data were completely anonymized prior to testing. All samples underwent an anonymization procedure in order to adhere to the regulations issued by the local Ethical Board (AVR-PPC P09, rev.2). The following four molecular methods were evaluated: Allplex™ Entero-DR Assay (Seegene Inc., Seoul, Korea), Amplidiag^®^ CARBAR+MCR (Mobidiag, Espoo, Finland), AusDiagnostics MT CRE EU Assay (AusDiagnostics, Mascot NSW, Australia), and EasyScreen™ ESBL/CPO Detection Kit (Genetic Signatures, Newtown NSW, Australia). Almost all samples were evaluated with the four techniques, with exceptions due to the lack of residual sample volume or test reagents.

#### 2.2.1. Allplex™ Entero-DR Assay (Seegene Inc., Seoul, Korea)

The rectal eSwabs (number of specimens tested = 1004) were processed using the Allplex™ Entero-DR Assay. Samples were vortexed and loaded in the Microlab STARlet IVD Liquid Handling Workstation (Hamilton Robotics, Reno, Nevada) with a Universal Extraction Kit (Seegene). DNA extraction was carried out with 200 µL of primary sample, and DNA was eluted in a volume of 100 µL. Five microliters of DNA extract was mixed with 20 µL of master mix, and qRT-PCR was performed using a CFX96 system (Bio-Rad, Hercules, CA, USA). All procedures were performed according to the manufacturer′s instructions. The test results were interpreted automatically and were presented using the Seegene Viewer software.

#### 2.2.2. Amplidiag^®^ CARBAR+MCR (Mobidiag, Espoo, Finland)

Automated nucleic acid extraction was performed with 200 µL of primary sample (number of specimens tested = 1015) using the GENEQUALITY X120 Pathogen kit (Mobidiag). An internal control was added during this step that served as a process control. Both the extraction and PCR setup were carried out on the GENEQUALITY X120, based on the STARlet platform (Hamilton Robotics, Cary, NC, USA). Up to 60 samples can be processed when the two analytical steps are paired. The RT-PCR amplification reaction was performed on a CFX96 thermal cycler (Bio-Rad). The assay was divided into three different multiplex reactions, each amplifying a specific group of target genes. The results were processed and reported by Amplidiag Analyzer software.

#### 2.2.3. AusDiagnostics MT CRE EU Assay (AusDiagnostics, Mascot NSW, Australia)

A total of 834 rectal eSwab specimens were processed using the AusDiagnostics MT CRE Assay. Since this platform does not have proprietary extraction technology, DNA was extracted using the same method as Allplex (5 µL of DNA). The AusDiagnostics MT CRE Assay is based on two different steps. The first step involves a short 15-cycle pre-amplification reaction using a mixture of primers homologous to each one of the 14 targets and two internal controls. The amplified product is then diluted into individual wells for the second step that is dedicated to target specific nested (located inside the sequence used for step one) amplification. The process is automated using the Easy-Plex liquid handling robotics system (AusDiagnostics). MT CRE Assay uses two different types of controls: one is a pan-bacterial target used to validate the sample quality (intrinsic control), and the second is an artificial DNA sequence used as reference for the semi-quantitation of the different targets (PCR control). The final and overall evaluation of the results is automatically achieved by MT-PCR Analysis software.

#### 2.2.4. EasyScreen™ ESBL/CPO Detection Kit (Genetic Signatures, Newtown NSW, Australia)

A total of 827 samples were processed using the EasyScreen™ ESBL/CPO Detection Kit. Sample preparation involved a patented system (3base™) for the conversion of the sample DNA into a synthetic nucleic acid with a three-base code. A 150 μL liquid specimen was mixed into a lysis buffer containing a bisulfite reagent, and then it was vortexed and incubated at 95 °C for 15 min; this step ensured lysis of the microorganisms and the universal modification of its nucleic acid into a three-base form, in which all cytosine bases are converted to thymine through a uracil intermediary. Nucleic acid extraction and PCR setup were performed on the GS1 Automation System, an instrument based on the Nimbus platform (Hamilton Robotics). Up to 91 samples can be processed within a single batch. The RT-PCR amplification reaction was performed on a CFX384 thermal cycler (Bio-Rad). Along with the target sequences, a universal bacterial sequence was detected as well; this should be endogenously present in all stool specimens and acts as an extraction control.

## 3. Results

A total of 1015 rectal surveillance swabs underwent evaluation by culture. The prevalence detected for KPC-producing Enterobacteriaceae was around 2,2% (22/1015), while the detection of MBLs and OXA-48 was infrequent (NDM-1 = 2 samples, VIM = 1 sample, OXA-48 = 2 samples).

The number of available tests was different from one molecular assay to another depending on the reagents made available for this study by the different diagnostic companies (Table 1). However, the analytical performance of every molecular assay was evaluated, and results were compared with those found by using the routine culture-based screening technique (Table 2).

The Allplex™ Entero-DR Assay yielded 982 valid results (42 positives) over 1004 processed swabs. Data collection was subject to automated analysis by Seegene Viewer software. It is of note that in 22 samples the intrinsic control amplification failed. Subsequently, the PCR control was spiked into the extracts of the 22 invalidated samples and was amplified to prove the functionality of the reaction mix. In all 22 samples, the PCR control signal was confirmed, and the presence of inhibitors was excluded.

The Amplidiag^®^ CarbaR+MCR kit yielded 983 valid results (30 positives) over 1015 processed specimens. The system excluded 32 swabs on the basis of the physical properties of the samples during the extraction step.

The AusDiagnostics MT CRE EU assay processed 837 DNA samples, which were previously extracted with the Universal Extraction Kit (Seegene). A total of 834 (24 positives) results were valid; in 3 cases both the intrinsic and PCR controls were not amplified, likely because of the presence of inhibitors in these samples.

The EasyScreen™ ESBL/CPO Detection Kit provided 787 valid results over 827 tested specimens (33 positives). The remaining 40 samples were excluded from analysis for lack of amplification due to the probable presence of inhibitors.

Every result obtained from each commercial kit was compared with both the routine reference method and the other three assays. A result was considered discordant if it was not in agreement with at least one other test taken into account, either culture or molecular methods. Discordant test results were further analyzed to differentiate them as follows:

blue (Figure 1) results were concordant with the routine screening test but conflicted with at least one other molecular assay;

red (Figure 1) results were in conflict with routine screening but agreed with at least one other molecular test; and

green (Figure 1) results were in conflict with both routine screening and all other molecular assays.

Discordant test results are illustrated in Figure 1.

Over 31 total discordant results, the Allplex Entero-DR assay showed 14 tests that were in agreement with routine screening (7 positive over 14 samples) but conflicted with at least one other molecular assay (Figure 1. blue). This assay identified 6 samples positive for VIM and, according to another molecular kit, 2 positive for OXA-48 and 1 positive for KPC. Those 9 samples were not detected by the routine method (Figure 1. red). Additionally, in 7 specimens, this test only gave a positive result for single targets including KPC (n=4), VIM (n=1), OXA-48 (n=1), and IMP (n=1). In one rectal swab, three genes were detected, but not confirmed by any other test, as follows: blaKPC, blaVIM, and blaIMP (Figure 1. green).

Eighteen Amplidiag^®^ CarbaR+MCR results were in agreement with the cultural screening out of 33 total discordant results (3 positive over 18 samples) (Figure 1. blue). Nine samples were not detected by routine screening but were detected according to at least one additional molecular method, of which 6 were VIMs and 2 were KPCs. In the remaining samples, this method did not show blaKPC, unlike routine screening, the Allplex Entero-DR assay, and the AusDiagnostics MT CRE EU assay (Figure 1. red). In 4 cases the detection of VIM was not confirmed by any other technique. In 2 cases the assay did not detect the KPC target, unlike all other tests (Figure 1. green).

Over 22 total discordant results, the AusDiagnostics MT CRE EU assay showed 17 tests that were in agreement with routine screening but conflicted with at least one other molecular assay (6 positive over 17 samples; Figure 1. blue). This assay identified 2 samples positive for VIM, 2 for OXA-48, and 1 for KPC, which were not detected by culture (Figure 1. red). The AusDiagnostics MT CRE EU assay did not show results in conflict with both routine screening and all other molecular assays.

Eighteen EasyScreen™ ESBL/CPO test results were in agreement with the cultural screening out of 28 total discordant results (2 positive over 18 samples) (Figure 1. blue). In 2 samples, bla OXA-48 detection was supported by 2 other molecular methods (Allplex Entero-DR and AusDiagnostics MT CRE EU) but not by culture (Figure 1. red). In 8 specimens the results conflicted with all other tests. In particular, in 3 cases the KPC target was not detected, and in 5 cases bla OXA-48 was detected only by this test (Figure 1. green).

## 4. Discussion

The limited therapeutic options and the high mortality associated with CRE infections impel us to put in place measures to minimize the chance of transmitting such infective agents from colonized to non-colonized subjects. Active surveillance to identify asymptomatic carriers is fundamental for proper care management and for an effective implementation of control measures. On one hand, screening plates represents a relatively low-cost method for isolating CRE from rectal swabs; on the other hand, molecular assays for the detection of clinically significant carbapenemase genes have become widely available in the last years. All patients are clinically evaluated in accordance with a CPE Control Care Checklist on admission to the healthcare setting. The patients included in one of the risk categories listed in this checklist are managed for up to 48 h as potential CRE carriers while waiting for the results of screening cultures to become available.

An internal investigation of the Management of the Greater Romagna Area Health Unit has estimated the average cost of personal protective equipment at 23 euros/48 h, including all medical devices required for the contact isolation period. The average cost of CRE screening based on plating the specimen for growth on selective media, followed by additional testing for isolate confirmation, is around 2 euros/sample. In comparison, the average cost of CRE screening based on molecular assays tested in this study was about ten times the price. Taking into account this expense, molecular tests could become cost-competitive, especially when the turnaround time is decreased dramatically from 48 to about 8 h, and caregivers have the chance to reconsider pre-emptive isolation and to scale back contact precautions within the same day of hospital admission. Furthermore, considering laboratory testing merely in terms of cost does not take into account the added expenses that derive from delayed results, and the costs of new rapid methods come with an overall reduction in hospital costs and improvements in patient care.

The four commercial molecular assays evaluated in this study did not require any modifications in the collection, transport, and storage of rectal swabs routinely processed by culture-based techniques. Loading collection tubes is convenient, practical, and it reduces errors in the pre-analytical phase (especially when associated with liquid handling workstations and barcoded materials) such as in the Allplex and Amplidiag platforms. The AusDiagnostics assay, on the other hand, does not provide a proprietary extraction system, leaving the choice of the most suitable solution to the laboratory. The sample-to-result instruments are handy since DNA extraction, amplification, and result interpretation phases are integrated into a single system; none of the assays evaluated provided this feature. Because of the structures of these tests, an additional operator intervention is required. Considering the importance of releasing lab results in the shortest possible time, a suitable assay should be easy for all laboratory personnel to use and available 24 h/day and 7 days/week. The EasyScreen protocol involves multiple manual steps and sample transfers, requires considerable attention to avoid sample mix-ups, and has a significantly longer hands-on time compared to other molecular techniques.

Molecular assays showed results concordant with the reference culture-based method in most cases. The AusDiagnostics MT CRE EU assay displayed the highest rate of concordance with the routine cultural screening assay.

Comparisons between results obtained with each commercial platform were only performed for discordant samples. Concordant test results between two or more molecular techniques, in the case of a negative reference test, should be considered as likely true positives given the superior sensitivity of the PCR-based approach [34,35]. The Allplex kit detected the highest number of carbapenemase genes, according to the other molecular assays, in the absence of growth of carbapenem-resistant colonies. This improved sensitivity is a key feature in the screening of potential carbapenemase-producing Enterobacteriaceae carriers; any false negative test result is especially dangerous and undermines the whole effort of the surveillance program. Amplidiag and EasyScreen were not able to detect resistance genes in samples that turned out to be positive for CPE in routine testing; this carries the risk of CRE spreading from colonized subjects to neighboring patients. On the other hand, reliable detection of CRE carriage is important to optimize public health care resources and to guide hospital infection control interventions. For this reason, it is recommended to restrict the number of false positives. Allplex and EasyScreen methods provided the highest number of positive results in conflict with both routine screening and all other molecular assays.

An additional advantage of molecular methods involves the detection of carbapenemase producers with low-level resistance, which are difficult to identify with conventional techniques (e.g., OXA-48 producers) [36,37]. Allplex, AusDiagnostics, and EasyScreen assays have shown the presence of the blaOXA48 gene in some rectal swabs, which were negative for routine methods.

Nucleic acid extraction and purification from fecal specimen suspensions are critical steps when the molecular approach is selected to diagnose CRE. The extraction control allows the monitoring of possible problems resulting from poor extraction yield, and the PCR control is designed to detect inhibition of the polymerase chain reaction. These potential issues could affect the efficiency of PCR and, consequently, the clinical accuracy of the results provided. The four commercial molecular assays evaluated in this study were supplied with both extraction and PCR controls. Lower amounts of amplifiable human DNA make the bacterial 16S rRNA assay a better candidate to control sample adequacy. A rectal swab quality control check based on visually ensuring the presence of fecal material does not represent an optimal solution in an automated laboratory workflow; in practice, this is time-consuming and unreliable. Allplex, AusDiagnostics, and EasyScreen methods provide intrinsic control in order to ensure sample adequacy.

Molecular methods have the advantage of directly detecting more than one target in a single test, thus expanding epidemiological surveillance capabilities. Active surveillance of rectal swabs could have a positive effect in implementing all evidence-based control measures to protect the health of the public. The assays evaluated in this study allow the characterization of clinically relevant antibiotic resistance genes from the most common healthcare-associated MDR bacteria. Genes associated not only with carbapenem resistance but also with colistin (*mcr-1*), vancomycin (*VanA* and *VanB*), and beta lactam resistance (e.g., CTX-M) have been detected in different specimens (Table 3).

The platforms evaluated in this study can perform an array of molecular diagnostic tests in the area of infectious diseases and also in other areas. Seegene’s platform can perform a wide variety of tests in infectious diseases (sexually transmitted diseases, respiratory tract and central nervous system infections, healthcare-associated infections, and surveillance), as a Thrombosis SNPs (single nucleotide polymorphisms) panel, and in detecting the BRAF mutation (V600E) characteristic of papillary thyroid carcinoma. Amplidiag^®^ is a family of diagnostic tests for the screening of gastrointestinal pathogens and antibiotic resistances. Ausdiagnostics has developed a large number of assays in the area of human infectious diseases (bacterial drug resistance, respiratory tract and central nervous system infections, enteric pathogens, HPV genotyping), veterinary infectious diseases, and food pathogens. Genetic Signatures offers a series of assays for the detection of enteric pathogens (bacteria, protozoa, and viruses).

The spread of CPE in the Emilia-Romagna region, infrequent until 2009, showed a clear increase in reports of individual cases and hospital epidemics caused by KPC-producing *Klebsiella pneumoniae*. Publication of regional guidelines for the control of Carbapenemase-producing Enterobacteriaceae in July 2011 led to the implementation of an active surveillance program in the Romagna area. In the following two years a reversal of the trend was observed, which was characterized by a 40% reduction in CPE carriers among total swabs processed and by a 35% reduction in CPE infections overall among Enterobacteriaceae isolated from clinical samples. The situation had drastically worsened in 2015 when the number of identified CPE infections more than doubled compared to the previous year. This change led to expanded screening involving a greater number of patients admitted into different wards. In 2018 there was a sharp reversal; hospitals experienced a halving of CPE carriers and a 30% decrease in CPE infections.

Among total isolated CPE, KPC-producing *Klebsiella pneumoniae* is the germ most frequently found in rectal swabs and in clinical specimens. Additionally, a sharp increase in the number of carbapenemase-producing *E. coli* has been observed. In 2016, the prevalence of carbapenemase-producing *E. coli* isolated in the Romagna area was 5,6% over the total 886 CPE strains isolated. This percentage reached 7,8% in 2017 and then increased up to 10,7% in 2018. This tendency confirms the data published by Del Bianco [38] and poses a serious threat to human health, bearing in mind that *E. coli* is one of the most common members of the intestinal microflora and one of the main pathogens responsible for a broad spectrum of invasive diseases [39,40]. Carbapenemase-producing *E. coli* have spread, and the accumulation of resistance genes has prompted us to undertake upgraded strategies for expediting the diagnosis of MDR Gram-negative bacteria and for enhancing infection prevention and control in healthcare settings.

Because of its highest agreement rate, both with routine screening and other molecular methods evaluated, AusDiagnostics MT CRE EU has proved to be the most reliable assay among the commercial kits investigated in this study. Additionally, this method provided the lowest number of invalid samples, and it is supplied with a double internal control system to monitor the influence of possible inhibitory activities and to ensure sample adequacy. It is noted that this platform does not have proprietary extraction technology, leaving the choice of extraction method to the laboratory depending on the existing workflow.

## Figures and Tables

**Figure 1 microorganisms-07-00704-f001:**
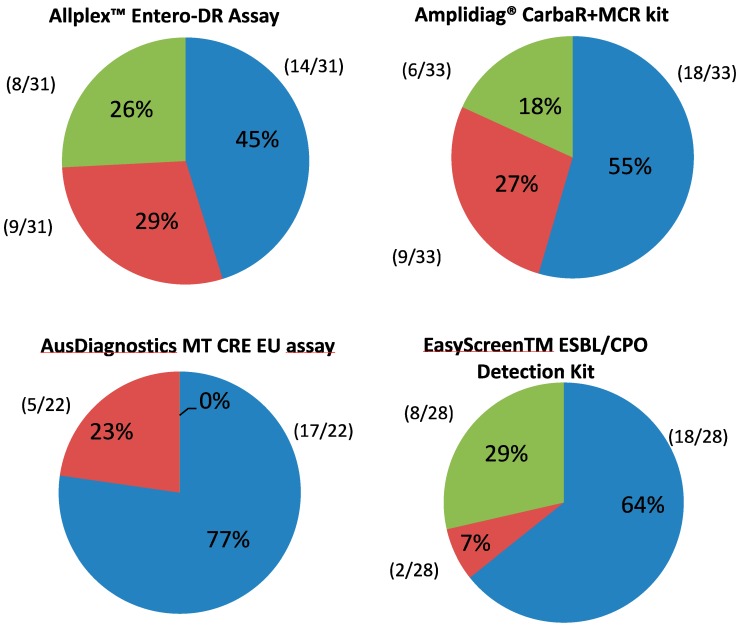
Analysis of discordant results. 
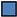
 Target concordant with routine screening but not in agreement with at least one other molecular test. 
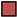
 Target not in agreement with routine screening but concordant with at least one other molecular test. 

 Target not in agreement with either routine screening or any other test.

**Table 1 microorganisms-07-00704-t001:** Detailed results of study specimens.

Assay		Routine Screening Tests		
	Assay Results	Positive	Negative	Total
Allplex Entero-DR assay	Positive	25	17	42
	Negative	0	940	940
	Total	25	957	982
Amplidiag CarbaR + MCR kit	Positive	20	11 ^a^	31 ^a^
	Negative	4	949	953
	Total	24	959	983
AusDiagnostics MT CRE EU assay	Positive	19	5	24
	Negative	0	810	810
	Total	19	815	834
EasyScreen ESBL/CPO Detection Kit	Positive	22	5	27
	Negative	0	751	751
	Total	22	756	778

^a^ A specimen was KPC-positive for routine screening, while with the Amplidiag kit, specimens were positive for KPC and VIM targets.

**Table 2 microorganisms-07-00704-t002:** Assay performance.

Assay	Sensitivity	Specificity	PPV	NPV	Overall % Agreement	Kappa Statistic
	(%[95%CI])	(%[95%CI])	(%[95%CI])	(%[95%CI])	(%[95%CI])	
Allplex	100	98.22	59.52	100	98,27	0.74
Entero-DR assay	(86.28–100)	(97.17–98.96)	(47.86–70.20)		(97.24–98.99)	
Amplidiag	83.33	98.85	64.52	99.58	98.48	0.72
CARBAR + MCR kit	(62.62–95.26)	(97.96–99.43)	(49.59–77.07)	(98.98–99.83)	(97.50–99.14)	
AusDiagnostics	100	99.39	79.17	100	99.4	0.88
MT CRE EU assay	(82.35–100)	(98.57–99.80)	(61.33–90.10)		(98.61–99.81)	
EasyScreen	100	99.34	81.48	100	99.36 (98.51–99.79)	0.89
ESBL/CPO Detection Kit	(84.56–100)	(98.46–99.78)	(64.75–91.33)			

**Table 3 microorganisms-07-00704-t003:** Main characteristics of four commercial molecular kits for detecting carbapenemase genes in rectal swabs.

	Allplex Entero-DR Assay	Amplidiag CarbaR + MCR Kit	Ausdiagnostics MT CRE EU Assay	EasyScreen ESBL/CPO Detection Kit
Sample throughput	up to 94 tests/batch	up to 64 tests/batch	24 up to 64 tests/batch	up to 80 tests/batch
Hands on time	45 min	1 h	20 min ^a^	3 h
Assay run time	4 h	5 h	2 h ^a^	6 h
Extraction control	yes	yes	yes	yes
PCR control	yes	yes	yes	yes
intrinsic control	yes	no	yes	yes
Other targets	vanA; vanB; CTX-M	AcOXA; MCR 1/2; Guiana extended-spectrum (GES)-CPO;	SME; OXA-23,51,58-like; CTX-M group 1 and group 9; GES	TEM; DHA; CTX-M; CMY SHV; OXA 23; 51-like
Traceability	yes	yes	Depending on DNA extraction system	no

^a^ Without extraction step.

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
