# Peer review of "Comparison of Four Commercial Screening Assays for the Detection of blaKPC, blaNDM, blaIMP, blaVIM, and blaOXA48 in Rectal Secretion Collected by Swabs"

_microorganisms, 2019, doi:10.3390/microorganisms7120704_

Round 1

Reviewer 1 Report

In the manuscript by Del Bianco and colleagues the authors aimed at comparing four commercially available screening assays as diagnostic tools in clinical settings. The present work is interesting for medical community and also for academics and researchers. Nevertheless, the manuscript has considerable weaknesses and presentation quality should be  improved in order to improve clarity.

There are typing errors, incorrect formating, different designations adopted in different parts of the manuscript, etc.

Specific comments

-Lines 55-56. This phrase is not clear

-Line 136: Since this platform.... The DNA of different samples was extracted using the same protocol or the DNA of the same samples was used for both kits.

Lines 162, 171, 206, 207: why the yellow?

Line 168: Table 1

The tables in general are not easy to read. In this table I think the invalids should be either removed or included in the total. Like this is only a confounding factor. 

Line 201: Figure1 does not have enough resolution

Line 204: Where is Fig 1.1.a? It looks like there is an issue with figures' identification please check not only this line but also lines 205/206, line 208, line 210, line 213, etc

Overall discussion must be improved.

In order to improve clarity the authors should consider including as supplementary table all the results obtained for all the samples.

Author Response

Point 1: Lines 55-56. This phrase is not clear

Response 1: The sentence has been reworded. MBLs definition has been provided at lines 51-52.

Point 2: Line 136: Since this platform.... The DNA of different samples was extracted using the same protocol or the DNA of the same samples was used for both kits.

Response 2: The DNA of the same samples was used for both kits, this was clarified on the text.

Point 3: Lines 162, 171, 206, 207: why the yellow?

Response 3: It was just an oversight. It was corrected.

- Line 168: Table 1

The tables in general are not easy to read. In this table I think the invalids should be either removed or included in the total. Like this is only a confounding factor. 

We removed the invalids from the table 1. They are explained properly and in full on the results.

- Line 201: Figure1 does not have enough resolution

The figure format has been changed.

- Line 204: Where is Fig 1.1.a? It looks like there is an issue with figures' identification please check not only this line but also lines 205/206, line 208, line 210, line 213, etc

The extensions .a .b .c refer to captions below the figure. In order to clear the question, we have replaced the letters (a, b, c) with colours (blue, red, green).

- Overall discussion must be improved.

We have deepened aspects relating to cost of molecular-based compared with culture-based testing and other uses of these platforms.

- In order to improve clarity the authors should consider including as supplementary table all the results obtained for all the samples.

We decided to provide as supplementary material a table presenting discordant results as opposed to all results as we believe the former is able to present data more clearly, while the latter would be dispersive and hard to read, without presenting further information

Reviewer 2 Report

Authors have done excellent job of comparing CRE related 4 assays.  Few suggestions are-

I. In table 3, it would be helpful to include row for per sample cost for chemicals and consumables. 

II. In discussion, authors can discuss any other uses of these platforms if so. 

Author Response

- In table 3, it would be helpful to include row for per sample cost for chemicals and consumables.

Considerable price differentials can be documented comparing different methods, for example culture-based vs molecular-based testing for screening, or immunochromatographic assays vs colorimetric assays for isolates confirmation. The commercial assays that have been evaluated in this study rely on high-multiplex technologies and high-throughput automation systems (automated DNA extraction and PCR setup, followed by detection by real-time PCR and automated data analysis). Sample cost for chemicals and consumables for all is about the same price and depends on business proposal (sales forecast and commercial strategies), making it difficult to make an estimate.

- In discussion, authors can discuss any other uses of these platforms if so.

 We expanded in the text alternative  applications for all the platforms discussed.

Round 2

Reviewer 1 Report

The quality of the manuscript was improved but the following aspects should be taken into account.

Lines 112-113: “The following four molecular methods were evaluated.” Either refer the 4 methods or re-write the sentence.

Table 1. It looks like one line is missing at the bottom.

Lines 191-195: In my opinion, it is not clear to what the authors are referring with the green blue and red because the number of the figure is placed in the end of the paragraph. Please rearrange this part of the text. Either mention the figure number in the beginning or quote the figure number after the colour.

The reference to the supplementary results is missing and must be included.

Author Response

The quality of the manuscript was improved but the following aspects should be taken into account.

Point 1: Lines 112-113: “The following four molecular methods were evaluated.” Either refer the 4 methods or re-write the sentence.

Response 1: This has been clarified in the present text:

The following four molecular methods were evaluated: Allplex™ Entero-DR Assay (Seegene Inc., Seoul, Korea), Amplidiag® CARBAR+MCR (Mobidiag, Espoo, Finland), AusDiagnostics MT CRE EU Assay (AusDiagnostics, Mascot NSW, Australia), EasyScreen™ ESBL/CPO Detection Kit (Genetic Signatures, Newtown NSW, Australia).

Point 2: Table 1. It looks like one line is missing at the bottom.

Response 2: We agree with this comment. One line should be included at the bottom.

Point 3: Lines 191-195: In my opinion, it is not clear to what the authors are referring with the green blue and red because the number of the figure is placed in the end of the paragraph. Please rearrange this part of the text. Either mention the figure number in the beginning or quote the figure number after the colour.

Response 3: This has been clarified in the present text:

A result was considered discordant if it was not in agreement with at least another test taken into account, both culture and molecular methods. The discordant test results were further analyzed to differentiate them as follows :

blue (Figure 1) the results which were concordant with the routine screening test, but in conflict with at least one other molecular assay;

red (Figure 1) the results which were in conflict with the routine screening, but in agreement with at least one other molecular test;

green (Figure 1) the results which were in conflict with both the routine screening and all other molecular assays.

The discordant test results are illustrated in Figure 1.

Point 4: The reference to the supplementary results is missing and must be included.

Response 4:

negative a (routine screening tests) = no growth

negative b (molecular methods) = no signal detected for the targets

empty cell c = test not performed

KPC + d = blaKPC-containing isolate on culture

KPC, NDM, VIM, IMP, OXA-48,  AcOXA, MCR 1/2, GES-CPO,  VanA, VanB, CTX-M, TEM, DHA, CTX-M, CMY, SHV, OXA 23, 51-like,  SME, OXA-23,51,58- like, CTX-M group 1 and group 9, GES e =  targets detected